# Evaluation of the B.strong Queensland Indigenous Health Worker Brief Intervention Training Program for Multiple Health Risk Behaviours

**DOI:** 10.3390/ijerph18084220

**Published:** 2021-04-16

**Authors:** Frances C. Cunningham, Majella G. Murphy, Grace Ward, Royden Fagan, Brian Arley, Peter H. d’Abbs

**Affiliations:** 1Wellbeing and Preventable Chronic Disease Division, Menzies School of Health Research, Level 10, East Tower, 410 Ann Street, Brisbane 4000, Australia; majella.murphy@menzies.edu.au (M.G.M.); royden.fagan@menzies.edu.au (R.F.); brian.arley@menzies.edu.au (B.A.); peter.dabbs@menzies.edu.au (P.H.d.); 2Diabetes Australia, 11 Finchley Street, Milton 4064, Australia; gward@diabetesaustralia.com.au

**Keywords:** program evaluation, brief intervention, workforce development, Indigenous, smoking, nutrition, physical activity, multiple health behaviours, primary health care

## Abstract

Queensland’s B.strong brief intervention training program was a complex intervention developed for Aboriginal and Torres Strait Islander health workers to assist clients address multiple health risks of smoking, poor nutrition and physical inactivity. This study evaluates program effectiveness by applying the Kirkpatrick four-level evaluation model: (1) Reaction, participants’ satisfaction; (2) Learning, changes in participants’ knowledge, confidence, attitudes, skills and usual practice; (3) Behaviour, application of learning to practice; and (4) Results, outcomes resulting from training. A retrospective analysis was conducted on data for respondents completing pre-training, post-workshop and follow-up surveys. Changes in domains such as training participant knowledge, confidence, attitudes, and practices between survey times were assessed using paired-samples t-tests. From 2017–2019, B.strong trained 1150 health professionals, reaching targets for workshop and online training. Findings showed statistically significant improvements from baseline to follow-up in: participants’ knowledge, confidence, and some attitudes to conducting brief interventions in each domain of smoking cessation, nutrition and physical activity; and in the frequency of participants providing client brief interventions in each of the three domains. There was a statistically significant improvement in frequency of participants providing brief interventions for multiple health behaviours at the same time from pre-workshop to follow-up. Indigenous Queenslander telephone counselling referrals for smoking cessation increased during the program period. B.strong improved practitioners’ capacity to deliver brief interventions addressing multiple health risks with Indigenous clients.

## 1. Introduction

### 1.1. Background

The decline in smoking prevalence for Aboriginal and Torres Strait Islander (hereafter referred to as “Indigenous” with acknowledgement of their distinct cultures) Australians shows the potential for improvement across multiple risk factors. From 1994 to 2019, smoking declined from 54.5% to 41.4% (compared with a decline from 23.5% to 14.4% for the non-Indigenous population) [1]. In the state of Queensland, with the Indigenous population estimated at 4.6% of the total population [2], life expectancy at birth in 2015–2017 for Indigenous males was 72 years and 76.4 years for females, compared to that for non-Indigenous Queenslanders, 79.8 years and 83.2 years, respectively [3]. Disparities in health status need to be considered in the wider context of Indigenous disadvantage associated with colonization, racism, poverty, inequitable access to the social determinants of health and exposure to environmental drivers of health risks [4]. Chronic disease accounts for 70% of the health gap between Indigenous and non-Indigenous Australians [5]. Addressing modifiable risk factors could prevent more than one-third of the burden of disease in Indigenous Australians [5]. The effect of these modifiable risk factors illustrates the potential for improved health outcomes for Indigenous Queenslanders through prevention. In addition, behavioural risk factors often co-occur in individuals, and they can have a synergistic effect in increasing cardiometabolic risk [6] and increasing morbidity and mortality [7,8,9]. It is important, therefore to reduce multiple health risks for Indigenous Queenslanders.

It is recognized that successful prevention efforts need strategies with multiple components, and they need to be implemented across different sectors, levels and settings. Brief interventions to address individual behavior change are one component. Brief interventions can lead to changes in health risk behaviour if they are delivered by practitioners trained in motivational interviewing; incorporate behavioural techniques, especially self-monitoring; are tailored to individual circumstances; and encourage the client to seek support from other people [10]. Combining brief interventions for multiple risk factors such as smoking, poor nutrition and physical inactivity can improve efficiency and has the potential to improve health outcomes, by providing a clinical framework to guide screening and intervention for multiple risk factors simultaneously [10]. The delivery of culturally appropriate brief interventions to address multiple risk factors is also consistent with the holistic, comprehensive approach to health care delivery in Australian Indigenous primary health care (PHC) services [11].

### 1.2. Background and Structure of the B.strong Program

Queensland Health initially funded Menzies School of Health Research (Menzies) to develop, implement and evaluate the Aboriginal and Torres Strait Islander Brief Intervention Training Program (B.strong) from October 2016 to August 2019 (A$2.3M), and subsequently extended the program to June 2020 (A$0.5M). Menzies developed, implemented and evaluated the B.strong program from 2017–2020. This study provides an evaluation of the operational program from June 2017 to August 2019.

As a professional capacity building program developed to address multiple health behavioural risks, the overall B.strong program meets definitional requirements of the United Kingdom’s Medical Research Council as a ‘complex intervention’ [12]. In providing guidance for evaluation, the Council describes complex interventions as interventions that contain several interacting components. Such complexity also includes: the number and difficulty of behaviours required by those delivering or receiving the program, and the number and variability of outcomes. B.strong was a multi-strategy intervention to provide Queensland’s Indigenous health and community workforce with the knowledge, skills and tools to deliver brief interventions in smoking cessation, nutrition and physical activity to promote healthy changes to their clients. B.strong training was provided across Queensland in all state health administration regions (Hospital and Health Services) to Indigenous health workers and other staff working with Indigenous clients in government and non-government health and community sectors. Training was also provided to staff working with Indigenous clients in non-health settings such as schools and correctional facilities. In Queensland, Indigenous people have access to PHC services provided by both the state government-operated PHC services and Indigenous community-controlled and operated PHC services [13]. In addition, general practitioners provide mainstream PHC services.

B.strong aimed to build staff capacity to assist their clients change their health risk behaviours, including multiple behavioural risks. Its objectives were to: (1) increase practitioner access to brief intervention training, (2) deliver more brief intervention services to Indigenous clients in primary and community care settings, (3) assess and refer more clients to early intervention programs and services, and (4) improve understanding and awareness of key risk factors for chronic disease in Indigenous communities in the longer term. B.strong provided culturally appropriate, evidence-based training (based on current Australian government guidelines) and resources. As shown in Figure 1 below, B.strong had multiple components. It included a one-day, face-to-face workshop (8 h duration), six online modules (each of two hours’ duration), practitioner tools and resources and client resources, along with trainee support through a help desk and a peer support network. 

The key features of B.strong are shown in Box 1 below. The development of the training and resource package was informed by social learning/social cognitive theory which emphasises the influence of environmental factors on willingness to change [14]. B.strong employed motivational interviewing: once key risk factors are identified, patients should be assessed for their readiness to change prior to motivational counselling and/or referral [15]. In Australian Indigenous PHC, this assessment is conducted during the annual Indigenous health check (Medicare Benefits Schedule Item 715). B.strong was developed on the foundation of the SmokeCheck Brief Intervention Training Program (2003–2008) [16] and the Aboriginal and Torres Strait Islander Nutrition and Physical Activity Brief Intervention Program (2003–2008) [17], both previously developed and delivered by Queensland Health. Findings from the SmokeCheck evaluation showed that it was effective in providing health workers with the practical skills, self-efficacy and confidence to conduct brief interventions with clients [16]. Workers found SmokeCheck feasible to use with clients and SmokeCheck brief interventions positively influenced smoking behavior in clients. The nutrition and physical activity brief intervention program was based on SmokeCheck. Subsequent Indigenous Australian brief intervention training programs were based on these innovative Queensland programs [18,19], but addressed the single risk factor of smoking.

Box 1Key features of the B.strong program.
From 2017–2020, for the overall B.strong program, over 1250 Indigenous health workers/practitioners and other health and community professionals received training in 107 one-day B.strong face-to-face workshops delivered by Indigenous training facilitators in Queensland. Training was also available for six online modules. B.strong content. Workshops: overview of cultural safety and security; overview of impact of smoking, poor nutrition and physical inactivity; behaviour change theory, motivational interviewing techniques and application; brief intervention delivery; reinforcing links to key public health messages; and information regarding relevant government and non-government programs and resources. Six online modules: (1) B.strong Introduction; (2) B.strong Essentials; (3) Quit for Health; (4) Eat for Health; (5) Move for Health; (6) B.strong in Practice. Practitioner resources: practitioner kit, handbooks, guides, community mapping tools. Client resources: Indigenous specific client brochures relating to stages-of-change.Based on current Australian guidelines for smoking cessation, nutrition and physical activity, ensuring resources are evidence-based.Based on the behavioural change theory (stages-of-change) approach [20,21]; uses the 5 A’s (Ask, Assess, Advise/Agree, Assist, Arrange) [22] as a health service framework for understanding the way in which prevention is implemented in PHC to address risk factors [23,24]; and the OARS approach (Open questions, Affirmation, Reflective listening, and Summary reflections) [25] as the core interactive skills of motivational interviewing.12 culturally-appropriate, evidence-based, plain language, client brochures were developed for staff trained in B.strong to use with their clients in three series: Quit for Health, Eat for Health, Move for Health.Provided opportunity for continuing professional development to meet ongoing professional registration requirements for B.strong health professional trainees.Implementation of B.strong aligned with current practice in the health service–brief interventions should become part of the routine client pathway and be recorded in the client’s electronic health record.Every client contact is an opportunity for a brief intervention [26,27].Ongoing monitoring, research and evaluation was conducted on the program.


A process of co-design and co-development was used to ensure that the B.strong training and resources were specifically developed to be culturally appropriate for the Queensland Indigenous population. B.strong development was informed by Menzies School of Health Research’s Guidelines for Engagement and Implementation for Community Research [28]. The importance of co-leadership and co-design in working with community was recognized throughout program development, implementation and evaluation, and included a range of processes. (1) An Indigenous Reference Group, including health professionals and health service consumers, provided input and feedback on all stages of development and design of the program, including program name, branding, content and program material, and online training resources, to ensure culturally safe and engaging program materials. (2) An Expert Advisory Group was established, with Indigenous leadership and representation, to guide development, review and finalization of B.strong training and resource materials. (3) Indigenous people and their communities were involved at all levels of participation and decision-making (e.g., piloting of the training program; feedback on program development, and ongoing program refinement from participants in training). (4) The program was developed to have the right ‘look and feel’ for the Indigenous health workforce (e.g., program branding was developed by an Indigenous graphic designer; co-developed animations and humour respectfully depicted brief intervention scenarios and associated skills; Indigenous voices delivered all online training; Indigenous health professionals provided feedback to further develop the program). (5) Culturally acceptable ways of implementing programs and doing evaluation research were used (e.g., involvement of an Indigenous evaluator in the development of the monitoring and evaluation plan; recruitment of two Indigenous training facilitators, one female and one male, an indigenous promotion and engagement officer, and other Indigenous staff on the B.strong team). Indigenous staff supported participants through a help desk and community of practice. (6) Strategic partnerships were built (e.g., with Queensland Aboriginal and Islander Health Council and with Apunipima Cape York Health Council). 

Brief interventions in PHC can range from three minutes of brief feedback and advice, to 15–30 min of brief counselling [25]. The B.strong brief interventions are intended to last for around 3–15 min, however the principles can be used for longer or recurrent intervention sessions should time allow. To enhance counselling success, practitioners may be able to capitalise on the “teachable moment”, estimated to occur in 10% of PHC office visits [29]. Such moments, which increase a client’s motivation to change, include acute medical events or pregnancy [29]. PHC workers are in a unique position to identify and intervene with clients presenting with behavioural risks. Health promotion and disease prevention play an important role in the work of PHC workers, who are often already engaged in implementing activities around screening and prevention including immunisation, and detection of high blood pressure, obesity, smoking and other health risk factors. PHC workers often have a trusted relationship with their clients and are a credible source of advice about health risks. For Indigenous clients, there is the importance of being able to discuss their lifestyle risks in a culturally safe environment [30]. In addition, through strong links with their communities, Indigenous health workers and practitioners can act as role models and can improve community knowledge and awareness of risks.

### 1.3. Previous Research on Brief Interventions

There has been considerable research on the use of brief interventions for smoking, nutrition and physical activity in the general population [31], however less research has been conducted with Indigenous populations, including the Australian Indigenous population, and little research has examined use of brief interventions for multiple health risks. A systematic review of tobacco cessation interventions for Indigenous populations world-wide found that facilitators for effective outcomes from brief interventions included the use of targeted interventions with appropriate resources and education, and use of culturally appropriate programs [32]. Challenges with conducting brief interventions included requiring a diverse workforce, and issues with long-term sustainability. Other reviews of brief interventions for smoking cessation in general populations found brief interventions had a positive impact on smoking cessation, and identified the importance of motivational interviewing in brief interventions [33,34].

A systematic review of dietary interventions for Australia’s Indigenous population found seven studies of nutrition education and promotion programs, however all were assessed as weak in quality [35]. Similarly, another review found a dearth of evaluations of physical activity interventions for Indigenous people in Australia and New Zealand [36]. A review examined behaviour-based intervention trials promoting fruit and vegetable intake in adults and children in the general population and in some minority groups [37]. It found that while behaviour-based interventions could increase fruit and vegetable intake, achieving and sustaining such intake at recommended levels across the population could not be achieved through behavioural interventions alone. The researchers flagged the need for combining these interventions with other approaches including social marketing, behavioural economics approaches and technology-based behaviour change models to achieve sustained goals. Another review evaluated brief intervention randomised controlled trials promoting change in eating habits in healthy adults [38]. It found that interventions providing education plus tailored or instructional components (e.g., feedback) were more effective than education alone, or non-tailored advice. Similarly, a review of randomised controlled trials supported the use of goal-setting and self-monitoring of behaviour for physical activity and healthy eating when counselling overweight and obese adults [39]. Other reviews support the effectiveness of interventions targeting diet and physical activity in sustaining weight loss [40,41]. For physical activity, reviews support the use of theory-driven, multi-component interventions [42], and also support the cost-effectiveness of brief interventions to promote physical activity in PHC and community settings [43]. 

The use of brief interventions to address multiple health behaviour change has been a largely understudied research area. Although many individuals engage in multiple interconnected health risk behaviours with the potential for negative health consequences, most health promotion research has addressed risk factors as categorically separate entities, and little is known about how to promote multiple health behaviour change [44,45]. There is increasing interest in understanding mechanisms shared across health behaviours that promote the co-occurrence of multiple health behaviours [45,46]. For example, recent research on cognitive connectionism supports the need to address patterns of associations between cognitive constructs and clusters of interrelated health behaviours [47]. A meta-analysis concluded that it is more effective to target smoking sequentially with other behaviours, rather than simultaneously [48], however more research is needed to inform this area [49]. One review of brief interventions for multiple risks in Indigenous populations in Australia, Canada and New Zealand identified the importance of addressing different intervention levels (individuals, community, population), and of including Indigenous participation and leadership in all stages of program development and implementation [50]. Johnson et al. (2014) examined the effects of co-action through three tailored interventions, showing that using a multiple-behaviour change approach made individuals 1.4 to 5 times more likely to make progress on a second behaviour [51].

This paper reports on the evaluation of the effectiveness of the B.strong brief intervention training program, using the New World Kirkpatrick Model as a framework [52]. The evaluation of the B.strong program provides the opportunity to learn from the implementation of an Indigenous brief intervention training program for multiple risk factors, compared with previous programs addressing single risk factors.

## 2. Materials and Methods

### 2.1. Evaluation Framework

We developed a Logic Model (Appendix A) to guide the evaluation [53]. Program evaluation was informed by the four-level New World Kirkpatrick Model for assessing adult professional training activities [52]. The Kirkpatrick Model is an internationally accepted standard model for evaluating the impact of adult training and development programs [54]. It has been widely applied in the health sector, including for evaluations of screening, brief intervention and referral to treatment (SBIRT) programs [55], and in Indigenous contexts, for example, in a systematic review of the impact of Indigenous health curricula on health professional learners [56]. 

Outcome indicators were used to measure how implementation of the B.strong program addressed the four levels of this model. Level 1, Reaction: satisfaction of participants with training; engagement of participants (degree to which they are actively involved in and contributing to the learning experience); relevance (degree to which training participants will have the opportunity to use or apply what they learned in training on the job). Level 2, Learning: changes in knowledge, confidence, attitudes, skills and usual practice of training participants in the delivery of smoking cessation, nutrition and physical activity brief interventions. Level 3, Behaviour: the degree to which participants apply what they learn during training when they are back on the job. Level 4, Results: the degree to which targeted outcomes occur as a result of the training and the support program (includes leading indicators, i.e., short-term observations and measurements suggesting that critical behaviours are on track to create a positive impact on desired results). We also assessed program impact: on access by Indigenous health workers and other health practitioners to brief intervention training across Queensland; on health service practice in provision of brief interventions to Indigenous clients; and on client referrals. 

The study authors are: F.C.C., a health services researcher who was the project lead at Menzies for the B.strong program, M.G.M., a psychologist who was the program manager, G.W., a diabetes educator, of Kamilaroi-Yuwaalaraay descent from South-West Queensland, who was a B.strong training facilitator; R.F., a health and community services trainer and smoking cessation counsellor, of Dja: bugaynji descent from North Queensland, who was a B.strong training facilitator; B.A., a community engagement and communications professional, of Torres Strait Islander (Tudu) descent from Daru Island, who was the B.strong community engagement officer; and P.H.d., a sociologist who conducted the data analysis for the evaluation. G.W., R.F. and B.A. are Indigenous researchers. F.C.C., M.G.M. and P.H.d. are non-Indigenous researchers, with research backgrounds in Indigenous health research and in program evaluation.

### 2.2. Data Collection

This study evaluated the delivery of B.strong from June 2017 to August 2019. For the 1150 participants who were trained during this period, data were collected through four surveys. Participants received a participant information sheet providing details about B.strong and there was an informed consent process. Participants were invited to complete three surveys: a pre-training survey, a post-workshop survey, and a follow-up survey, and there was one manager/supervisor survey. The paper-based pre-training survey was completed by training participants at the commencement of the face-to-face workshop, and an online version was available for online module only participants. The survey included demographic items (gender, age, Indigenous status, educational level, and time working in their current position), and items addressing self-assessed knowledge, confidence, attitudes, skills and usual practice in relation to the conduct of brief interventions. The paper-based post-workshop survey included the same items on knowledge, confidence, attitudes, as well as satisfaction with the workshop. The online follow-up survey was completed via Survey Monkey approximately three months following training. The follow-up survey included the same items on knowledge, confidence, attitudes, skills, as well as usual practice, satisfaction with the online modules, and feedback on B.strong training. Non-respondents to the online follow-up survey were followed up with several online reminders. The manager/supervisor survey was sent to managers and supervisors of participants three to six months following their training (however as the response rate was low (*n* = 9, 11% response rate), findings are not reported in this study). The surveys were piloted through two pilots of B.strong training that were held in an urban government-operated Indigenous PHC centre (*n* = 15 participants), and in a regional community-controlled Indigenous PHC centre (*n* = 15 participants). The pilot data are not included in this study.

Queensland Quitline provided quarterly data reports on self-generated and third-party referrals for Queensland Indigenous clients to assist with assessing the impact of B.strong training on trends in smoking cessation referrals for telephone counselling. Similar statewide referral data for nutrition and physical activity were not available, however clients could be referred either within their own service or externally for additional nutrition and physical activity support.

### 2.3. Statistical Analysis

Statistical analyses were performed using IBM SPSS version 25 (IBM, Armonk, NY, USA). Descriptive statistics were used to compute frequencies and percentages. Changes between pre-workshop and post-workshop survey scores, and between post-workshop and 3-month follow-up scores, were assessed using paired-samples t-tests. Although we are aware of continuing controversy regarding the applicability of parametric tests such as these to Likert scales, we note Norman’s [57] demonstration of their robustness and suitability for these data, and Sullivan and Artino’s [58] endorsement of their use in preference to non-parametric tests.

## 3. Results

### 3.1. Overall Training Targets

The B.strong program exceeded the training goal of 1100 workshop participants set by Queensland Health. From program commencement to August 2019, 1150 participants were trained (1131 workshops/online modules completers; 19 online modules only completers). Ninety-eight workshops were conducted across all of the 15 Queensland Hospital and Health Service geographic regions, compared with the target of 64 workshops. In total, 215 organisations had staff who completed B.strong training. Compared with the target of 360 for online module completion, there were 361 completers, a majority of whom had also completed the workshop training. Of these, only 19 opted to do just the online module training. Of note, in spite of equal promotion for the online training as for the workshop training, take-up was relatively poor, and there was no pattern of higher take-up of online modules from remote or regional areas. 

### 3.2. Participant Characteristics

For the data analysis, data were included for the main B.strong operational program. The analysis includes data on workshop participants trained until July 2019 and online module participant data to August 2019. Data were available for 1079 of the 1131 workshop participants (95% response rate), however, as indicated in our reporting below, the number of respondents varies across the surveys and across the different data items. Table 1 shows the demographic details for 1079 workshop participants, 62% of whom identified as being Aboriginal and/or Torres Strait Islander attendees. As Table 1 shows, participants were more likely to be female, aged 40 years and over, to have post-secondary qualifications, to have worked in their current position for at least two years, and to work at government-operated health services. 

### 3.3. Participants’ Satisfaction

Kirkpatrick Level 1, Reaction, addresses the participants’ satisfaction with the training, their engagement in it, and the perceived relevance of the training to their work. Participants provided positive feedback on the content and delivery of the workshops. (For this survey question, the wording of some items was changed during project implementation. We report here on responses to the later version (where *n* = 612)). There was a very high level of agreement in the post-workshop survey that the B.strong workshop objectives were clear to participants (98.5%), that the content was relevant to their job (92.8%), and that the workshop activities gave them sufficient practice and feedback on doing brief interventions (95.8%). Importantly, 97.7% reported that the workshops provided a culturally safe place for learning. When participants were asked about additional topics for B.strong training in the post-workshop survey, they most frequently requested inclusion of alcohol and other drugs.

There was also very positive feedback on the online modules. There were 361 completers of the online modules, however response rates varied across questions relating to the modules in the follow-up survey. Ninety-seven per cent of 229 respondents (63% response rate) found the modules interesting/very interesting, while 78% (*n* = 183 respondents) reported they were ‘very satisfied’ or ‘extremely satisfied’. A total of 80% (*n* = 186 respondents) would be ‘very likely’/‘extremely likely’ to recommend the modules to a work colleague. There were ratings of above 88% agreement from respondents (*n* = 225) across all areas of feedback on the modules, with the highest ratings for: ‘clear objectives’, ‘relevant quizzes’, ‘animations assisted with modelling brief intervention skills’, and ‘case study activities assisted learning’. Further, 90% (*n* = 225 respondents) found the module training relevant to their work.

### 3.4. Changes in Participants’ Knowledge, Confidence, Attitudes, Skills and Usual Practice

Kirkpatrick Level 2 indicators address learnings from training. Results are presented on the impact of B.strong training on changes in participant knowledge, confidence and attitudes to providing brief interventions to clients, in addition to its impact on the skills and usual practice of participants. Questions used 5-item Likert scales for responses. 

In the pre-training, post-workshop and follow-up surveys, participants were asked the question: “How do you rate your current knowledge of smoking, nutrition and physical activity and their impact on chronic illness? (e.g., heart disease, cancer, low birthweight, etc.)”. As there was a change in this question in the survey from April 2018, only data from April 2018 to August 2019 are used in this analysis. As shown in Table 2, we compare data pre- and post-workshop. To assess the statistical significance of changes over the three surveys in participants’ knowledge of the effects of smoking, nutrition and physical activity, paired-samples t-tests were used to compare means. For these tests, only participants who took part in the two surveys were included. Improvements were significant in all three domains (smoking, nutrition and physical activity) of knowledge (*p* < 0.001). Comparisons between post-workshop scores and scores at 3-month follow-up were limited to the smaller numbers participating in the follow-up surveys (*n* = 142 pairs; 12.5% response rate): they revealed small but non-significant declines at 3-months in self-assessed knowledge levels. However, scores at 3-month follow-up were significantly higher than pre-workshop scores in all three knowledge fields. Details of these trends are shown in Appendix A (comparisons between post-workshop and 3-month follow-up) and Appendix A (comparisons between pre-workshop and 3-month follow up).

There were two sets of questions relating to confidence: (1) “How confident are you in talking with your clients about how smoking/nutrition/physical activity may affect their health?” and (2) “How confident are you in assessing your client’s readiness to: quit smoking/improve nutrition/increase physical activity?”. Pre- and post-workshop scores are compared in Table 2. As Table 2 shows, improvements were statistically significant (*p* < 0.001) in all three health domains on both questions. Between post-workshop and 3-month follow-up, scores declined, but in most cases the decline was non-significant (Appendix A). Improvements between pre-workshop and 3-month follow-up were statistically significant (*p* < 0 001) in all domains of both indicators (Appendix A), suggesting that the positive impact of the workshop on participants’ confidence had been sustained. 

We also examined the impact of B.strong training on the attitudes of participants to providing brief interventions to clients, by asking participants to respond to three 5-point Likert type questions: (1) *“I have a clear idea of my responsibilities in helping clients with health behaviour change”*; (2) *“I feel there is little I can do to help clients change their health behaviour”*; (3) *“I feel uncomfortable asking clients about their health behaviours”*. Pre- and post-workshop responses are summarised in Table 2. As the table shows, changes in participants’ scores on all three items towards more positive attitudes and away from negative attitudes were statistically significant (*p* < 0 001). Changes in scores between post-workshop and 3-month follow-up were minimal and non-significant (Appendix A). However, comparison of pre-workshop scores with 3-month follow up scores revealed that the shift towards more positive attitudes in questions (2) and (3) was not statistically significant, suggesting a weakening of impact (Appendix A). 

The impact of B.strong training on participants’ practices was assessed through responses to two questions reported here: (1) *“How often do/will you ask your clients about their smoking/nutrition/physical activity?”* and (2) *“How often do/will you provide brief interventions to clients relating to smoking/nutrition/physical activity?”*. As Table 2 shows, responses demonstrated statistically significant improvements in practice in all three domains. Despite evidence of weakening impact between post-workshop and the 3-month follow-up (Appendix A), comparison between pre-workshop scores and 3-month follow-up scores indicated that the recorded improvements remained statistically significant (*p* < 0.05) (Appendix A).

The frequency with which participants reported asking clients about multiple health behaviours in a single presentation increased from 48.9% saying they did so often or always pre-workshop (*n* = 489) to 68.4% reporting doing so at the 3-month follow-up (*n* = 155). A paired t-test comparing scores on the two surveys, though limited to a smaller number of pairs (*n* = 86 pairs), showed that the improvement was significant (t = 2.536, *p* = 0.013).

### 3.5. Application of Learning to Practice

Kirkpatrick Level 3, Behaviour, addresses the degree to which participants have applied what they learned during training when they are back on the job. In the post-workshop survey, participants were asked what they would do differently in their own practice, or health setting, as a result of attending the B.strong workshop. The majority of participants (92%, *n* = 742/809) reported at least one or a number of things they would do differently in their own practice. Key areas included: the increased use of brief interventions, increased use of motivational interviewing, and their use of B.strong resources and tools (including use of stages-of-change, 5As and OARS). These findings were sustained in the follow-up survey, with 83% of respondents (*n* = 290) reporting the B.strong training led to similar changes in their practice when providing brief interventions.

### 3.6. Results of Training

Kirkpatrick Level 4, Results, covers the degree to which targeted outcomes occur as a result of the training and the support program. It includes short-term observations and measurements suggesting that critical behaviours are on track to create a positive impact on desired results. There was improvement in the frequency of trainees providing brief interventions to clients from pre-workshop to 3-month follow-up in all three domains. A paired t-test comparing scores on the two surveys, though limited to a smaller number of pairs (*n* = 86 pairs) showed that the improvement was significant in all three domains (smoking: t = 4.424, *p* = 0.000; nutrition: t = 4.558, *p* = 0.000; physical activity: t = 3.820, *p* = 0.000) (Appendix A).

Participants (*n* = 163) reported at 3-month follow-up on what they recorded in their service’s client records since taking part in B.strong training. The highest proportion (67%) recorded the client’s stage of readiness for behaviour change in smoking, compared with nutrition (66%) and physical activity (62%). There was a range of from 64–69% in the proportion of participants recording key areas for smoking, nutrition and physical activity brief interventions, such as: the client directed goals and monitoring tasks, the resources shared with the client, the referral details, and follow-up action and next appointment. Reflecting the take-up of the program post-training, in addition to the initial B.strong starter kits provided to trainees, the program distributed an additional 27,000 of each of the 12 client brochures to organisations participating in B.strong training.

Although data were not collected from Indigenous clients or from communities, we note that Indigenous health workers have a special role in their health services as a conduit back to their communities. Participant feedback in surveys reflected their perceptions of the benefit of B.strong for their clients and communities. For example, one participant stated:


*“It was a great workshop. I can assist my community to make better choices.”*
[Indigenous Social and Emotional Wellbeing Officer, Charleville, Queensland. Post-Workshop Survey]

Similarly, another participant conveyed: 


*“The program was great. I am taking a lot of new knowledge from this program that I can put into practice in my job to broaden skill sets and help clients at my fullest potential.”*
[Indigenous Health Promotion Officer, Warwick, Queensland. Post-Workshop Survey]

There was a significant increase in referrals for Indigenous clients to Queensland Quitline for telephone counselling on smoking cessation from 2053 referrals in FY2015/16 to 3479 in FY2018/19. This increase can likely be attributed to the impact of the B.strong program, in addition to promotional activities employed directly by the Quitline service to increase referrals for Indigenous clients. 

## 4. Discussion

### 4.1. Did B.strong Meet Its Objectives?

In this section we explore evaluation findings in the context of the extant literature and identify key lessons learnt that may help other population-wide, complex interventions on brief interventions. The purpose of the B.strong program was to train Indigenous health workers and other health and community professions in brief interventions to address multiple health risks in smoking, nutrition and physical activity with their Indigenous clients in Queensland. 

Overall, B.strong met its key objectives. (1) *Increase practitioner access to brief intervention training.* From 2017–2019, B.strong increased practitioner access to brief intervention training through training 1150 Queensland Indigenous health practitioners and other health and community professionals. (2) *Deliver more brief intervention services to Indigenous clients in primary and community care settings.* There were statistically significant improvements from pre- to post-workshop, and from pre-workshop to 3-month follow-up, in the frequency of participants asking clients about their smoking, nutrition and physical activity behaviour. Similarly, there were statistically significant improvements from pre- to post-workshop, and from pre-workshop to 3-month follow-up, in the frequency of participants providing brief interventions to their clients for all three domains. Provision of brief interventions for multiple health behaviours in a single presentation increased from 48.9% of participants stating they did so often or always pre-training to 68.4% at 3-month follow-up. (3) *Assess and refer more clients to early intervention programs and services.* There was an increase in referrals of Indigenous clients for smoking cessation counselling from July 2015, preceding B.strong implementation, to June 2019. (4) *Improve understanding and awareness of key risk factors for chronic disease in Indigenous communities in the longer term.* This was a longer-term objective, and although our evaluation did not collect data at the community-level to evaluate this, there was considerable survey feedback from participants on their perceptions of the benefit of B.strong training in assisting them to improve risk factors for their clients and their communities.

### 4.2. B.strong and the Kirkpatrick Indicators

Findings from the evaluation of B.strong in relation to the Kirkpatrick New World Model show that for Level 1, Reaction, there were high levels of trainee satisfaction for both the workshops and the online training.

For Level 2, Learning, study results provide evidence of program impact on improving the perceived knowledge, confidence, attitudes, skills and usual practice of Indigenous health workers and other health and community trainees in conducting brief interventions in smoking cessation, nutrition and physical activity. The increase in trainee capacity in these areas from comparatively low pre- training levels, and the sustaining of this increase at 3 months following training and returning to work with clients, highlight the benefit of B.strong training. In contrast to the overall improvements in these indicators, there was a weakening of impact from pre-workshop to 3-month follow-up for two of the three attitude questions. This might be explained by possible deficits at the health service level in workplace supervision and support for brief interventions, internal peer mentoring, and embedding of implementation targets as part of routine practice, along with monitoring and feedback to staff. However, the overall improvement in knowledge, confidence, attitudes, and skills in brief interventions of trainees was a prerequisite to their subsequent increased delivery of brief interventions.

Our findings are consistent with findings from the evaluation of the Queensland SmokeCheck program (2005–2006), a brief intervention training program for smoking cessation for Indigenous health workers [16]. That evaluation found increases for health worker participants in their self-efficacy, role legitimacy and confidence in discussing smoking cessation with clients. Similarly, the evaluation of the New South Wales SmokeCheck program (2007–2008), a brief intervention training program for smoking cessation for Indigenous health workers, found an increase in the confidence and attitudes of Indigenous health worker trainees towards conducting smoking cessation brief interventions with clients [18,59]. Findings from the evaluation of the Quitskills program (2012–2016), a brief intervention training program for Indigenous health workers across Australia, also found an increase in confidence, in knowledge and skills to address smoking for trainees [19]. In the latter study, increased self-confidence was associated with health professionals’ referrals of Indigenous smokers to smoking interventions. 

B.strong evaluation findings on the increase in trainees’ provision of brief interventions to clients following training are similar to findings from a Cochrane review of 17 trials which found that tobacco-related training programs helped health professionals to identify smokers and increased the number of people who quit smoking [32]. These programs also increased the number of people offered advice and support for quitting by health professionals. Our findings are consistent with findings from previous evaluations of Indigenous Australian SmokeCheck brief intervention training programs [16,59]. However, the Queensland evaluation referred only to the proportion of trainees using the program after training, rather than to their explicit use of brief interventions with clients. Compared with these previous Indigenous Australian programs which addressed the single risk factor of smoking, the B.strong program differed in addressing the multiple risk factors of smoking, poor nutrition and physical inactivity. Hence, the B.strong program was novel in demonstrating the capacity of a brief intervention training program to train participants in the delivery of brief interventions to Indigenous clients for multiple health risk factors. This is an important finding as a survey by Noble et al. [60] of clients of a New South Wales Indigenous PHC service identified that strategies addressing multiple health behaviour changes were likely to be acceptable to clients, particularly with support from health professionals, and involving family members. In addition, their research supported allowing for flexibility in the choice of initial target behaviour, timing of changes, and the format of support provided.

For Level 3, Behaviour, the majority of participants reported at least one or a number of things that they would do differently as improvements in their own practice of brief interventions following training, and this was implemented into their practice, as reported at 3-month follow-up. An evaluation of a Brazilian brief intervention training program also found that trainees applied the training directly in their practice [61]. 

In relation to Level 4, Results, through delivering 98 workshops and training 1150 participants, B.strong exceeded the delivery targets set by Queensland Health. In terms of attaining the major program objective, as also discussed under Level 2 above, there was a statistically significant increase from pre-workshop to 3-month follow up in the frequency of participants providing brief interventions for smoking, for nutrition and for physical activity. This finding is consistent with findings from an Australian randomised controlled trial of training of generalist community nurses where trained intervention group nurses reported assessing physical activity, weight and nutrition more frequently than controls, as well as providing more brief interventions for physical activity, weight management and smoking cessation [62]. Importantly, B.strong also had an impact on increasing the frequency of participants often or always conducting brief interventions for multiple health behaviours in a single presentation from pre-workshop (48.9%) to 3-month follow-up (68.4%). B.strong participants perceived that the program was beneficial for their clients and for their community. There was an increase in smoking referrals for Indigenous Queensland clients over the B.strong operational period, which may be associated with the impact of the B.strong program. 

### 4.3. A Complex Intervention–Effectiveness and Sustainability

The B.strong program was a complex intervention which included a number of components. Evaluations of brief intervention training programs for Indigenous health workers have identified the importance of combining training with other evidence-based strategies in multi-component interventions to enhance impact [63,64]. Nilsen and colleagues summarise several decades of research on alcohol brief interventions showing that multifaceted clinical interventions tend to be more effective than single interventions because they address multiple barriers to implementation [65,66]. Such evidence shows that passive approaches such as mailed educational materials, attending lectures, or doing online modules are generally ineffective and are unlikely to result in behaviour change when used alone. Active approaches incorporating skills-based behaviour rehearsal and feedback are more likely to be effective, but are also generally more expensive upfront. Unfortunately, passive approaches are the most common strategy adopted by researchers, professional bodies, and health care organisations. The findings on the significant increase in practitioners addressing multiple health risks of clients in a single presentation following training are promising given the added urgency of addressing such risks during the COVID-19 pandemic. This was reinforced by recent cardiovascular prevention advice from the United Kingdom highlighting the imperative of taking a holistic approach, and not treating risk factors in isolation [67]. With diminished prevention opportunities in the COVID-19 environment, it is especially important to make every contact in PHC an opportunity to ask about multiple lifestyle factors.

It is important to continue to provide B.strong training for new staff, given high workforce turnover, and refresher training and support for existing trainees to offset the decline in some of the indicators from post-workshop to 3-month follow-up. B.strong should be broadened beyond the Queensland Health program requirements to include alcohol and other drugs, so that all SNAP (smoking, nutrition, alcohol and physical activity) risk factors are addressed. To support the longer-term sustainability of the program, it will be essential for the B.strong brief intervention approach to be integrated into the routine clinical practice of all health service staff providing PHC to Indigenous clients [59,68,69,70]. This includes integration of brief intervention details into the electronic health record [71,72,73,74]. PHC service clinical protocols should align brief interventions with their practice workflow, and clearly identify the roles of Indigenous health workers and other interdisciplinary team members [71,75] in relation to brief interventions conducted opportunistically, as well as during the billable annual client health check. Easy-to-use system-level solutions that have electronic point-of-delivery reminders and decision support to facilitate coordination in PHC could be applied [10,76]. Finally, the positive evaluation findings on the B.strong program provide evidence for the national adoption in Australia of the B.strong program, or similar programs, to build the capacity of frontline practitioners to maximise opportunities to assist Indigenous clients in addressing multiple health risks.

### 4.4. Study Limitations

As the B.strong program was implemented as an operational program with government-specified annual targets for training delivery in each region, it was not feasible to conduct a cluster randomised control trial, or to employ a quasi-experimental design with a wait-list. It was also outside the scope of this evaluation to collect client-level data which would have provided information on the impact of the program on changes to their health risk behaviours. We note that most of the evaluation data were based on self-reported participant data rather than objective measures, and there could be the risk of responder bias or social desirability bias. As there was a relatively low level of participation in the 3-month follow-up survey, the follow-up results may need to be interpreted with caution. Nevertheless, there are important lessons from the pre- and post- training evaluation of this large-scale, state-wide program.

## 5. Conclusions

The B.strong program was developed as an evidence-based, culturally appropriate brief intervention training program to address the multiple health risks of smoking, poor nutrition and physical inactivity. Overall, the evaluation of the B.strong program shows that it met its key objectives in building staff capacity to assist their Indigenous clients change their identified health risk behaviours. The evaluation findings provide strong support for the effectiveness of an Indigenous brief intervention training program for building staff capacity to implement brief interventions to address multiple health risk behaviours. Study findings also provide further support for the employment of an active complex intervention approach in the brief intervention training program, rather than use of a single intervention. This study contributes to the limited research to date on building Australian Indigenous staff capacity in brief interventions, especially for multiple health risks, and the evaluation of the large operational B.strong program adds to the international literature on preventive programs for individual health risk behavior change.

## Figures and Tables

**Figure 1 ijerph-18-04220-f001:**
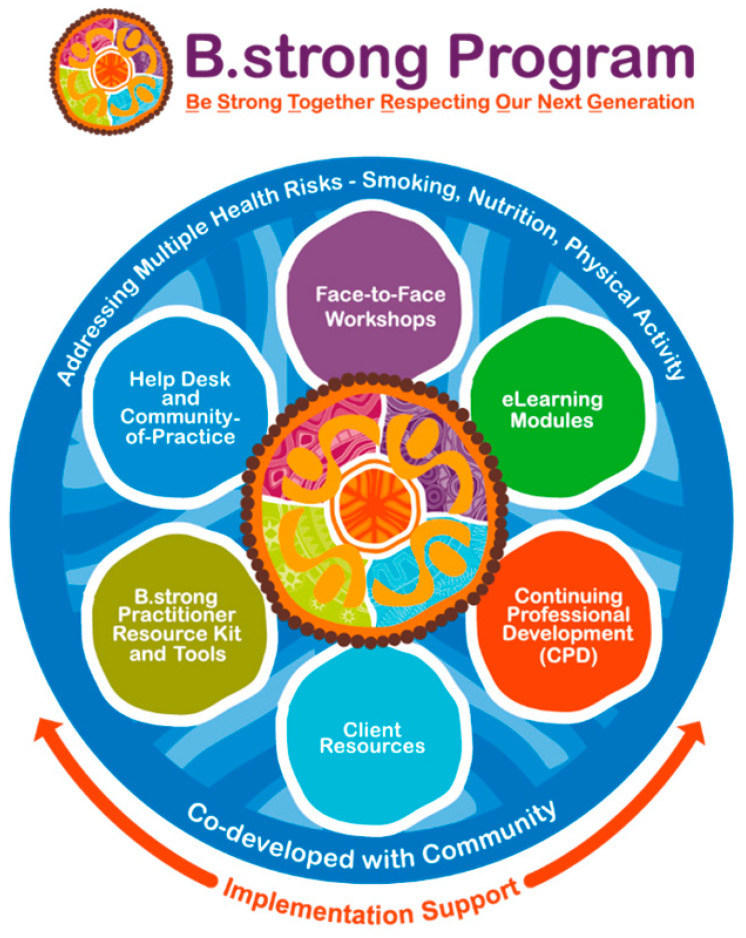
Components of the B.strong program.

**Table 1 ijerph-18-04220-t001:** Workshop participant demographics, frequencies and percentages (June 2017 to July 2019).

Demographic Variable	Categories	*n*	%
**Gender**	Female	858	79.5
	Male	213	19.7
	Other	2	0.2
	N.S. ^1^	6	0.6
	**Total**	**1079**	100.0
**Age-group**	Less than 25 years	105	9.7
	Between 25 and 40 years	382	34.4
	40 years or over	583	54.0
	N.S.	9	0.8
	**Total**	**1079**	100.0
**Indigenous status**	Aboriginal	498	46.2
	Both Aboriginal & Torres Strait Islander	81	7.5
	Torres Strait Islander	85	7.9
	Not Aboriginal or Torres Strait Islander	412	38.2
	N.S.	3	0.3
	**Total**	**1079**	100.0
**Education**	Postgraduate Degree	180	16.7
	Undergraduate Degree or equivalent	219	20.3
	Diploma/Advanced Diploma	222	20.6
	Certificate I, II, III or IV	334	31.0
	Year 12	51	4.7
	Below Year 12	41	3.8
	Other	7	0.6
	N.S.	25	2.3
	**Total**	**1079**	100.0
**Time working in role**	Less than 6 months	233	21.6
	6 months to 2 years	317	29.4
	More than 2 years	515	47.7
	N.S.	14	1.3
	**Total**	**1079**	100.0
**Organisation type**	Community-controlled health service	380	35.2
	Government-operated primary health care centre	84	7.8
	Private practice primary health care centre	11	1.0
	Community care centre	132	12.2
	Hospital and Health Service	293	27.2
	Educational facility	59	5.5
	Correctional facility	36	3.3
	Other	80	7.4
	N.S.	4	0.4
	**Total**	**1079**	100.0

^1^ N.S.: not stated.

**Table 2 ijerph-18-04220-t002:** Pre- and post-workshop comparisons.

Domain	*N* of Pairs	Pre WS ^1^ Mean (SD ^2^)	Post WS Mean (SD)	Mean Diff. (95% CI ^3^)	t	*p* Value
**Knowledge: how do you rate your knowledge of the impact on chronic illness of:**						
smoking	609	3.53 (0.935)	4.14 (0.718)	0.606 (0.539–0.673)	17.710	0.000
nutrition	607	3.53 (0.898)	4.14 (0.716)	0.608 (0.543–0.672)	18.512	0.000
physical activity	606	3.65 (0.882)	4.18 (0.708)	0.535 (0.472–0.597)	16.767	0.000
**Confidence (1): How confident are you in talking with your clients about their:**						
smoking	1004	3.44 (1.118)	4.06 (0.834)	0.621 (0.559–0.682)	19.826	0.000
nutrition	1005	3.49 (1.074)	4.1 (0.807)	0.614 (0.554–0.674)	19.964	0.000
physical activity	1004	3.53 (1.054)	4.12 (0.796)		19.146	0.000
**Confidence (2): How confident are you in assessing your clients’ readiness to:**						
quit smoking	998	3.06 (1.099)	4.01 (0.837)	0.951 (0.884–1.018)	27.939	0.000
improve nutrition	1001	3.17 (1.047)	4.06 (0.805)	0.883 (0.820–0.946)	27.566	0.000
increase physical activity	1000	3.21 (1.056)	4.06 (0.804)	0.853 (0.789–0.917)	26.219	0.000
**Attitudes** **:**						
Participants have a clear idea of their responsibilities in helping clients with health behaviour change	600	4.08 (0.714)	4.47 (0.608)	0.39 (0.332–0.448)	13.306	0.000
Participants feel there is little they can do to help clients change their health behaviours	601	2.41 (1.045)	2.26 (1.291)	−0.156 (−0.257–−0.056)	−3.068	0.002
Participants feel uncomfortable asking clients about their health behaviours	598	2.54 (1.117)	2.38 (1.275)	−0.157 (−0.263–−0.052)	−2.922	0.004
**Usual practices (1): how often do you ask clients about their:**						
smoking habits	444	3.76 (1.021)	4.32 (0.6770)	0.556 (0.474–0.638)	13.349	0.000
nutrition habits	445	2.09 (0.521)	2.46 (0.504)	0.553 (0.476–0.629)	14.187	0.000
physical activity	445	2.11 (0.499)	2.48 (0.505)	0.573 (0.495–0.651)	14.493	0.000
**Usual practices (2): How often do you provide brief interventions to clients relating to:**						
smoking	443	2.05 (0.565)	2.43 (0.536)	0.384 (0.329–0.439)	13.739	0.000
nutrition	444	2.09 (0.521)	2.46 (0.504)	0.374 (0.320–0.428)	13.541	0.000
physical activity	443	2.11 (0.499)	2.48 (0.505)	0.372 (0.320–0.425)	13.853	0.000

^1^ WS: workshop; ^2^ SD: standard deviation; ^3^ CI: confidence interval.

## Data Availability

Data are not publicly available due to restrictions, e.g., privacy considerations for the population of participants, and also because of service agreement requirements with the funder, Queensland Health.

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
