# Peer review of "Evaluation of the B.strong Queensland Indigenous Health Worker Brief Intervention Training Program for Multiple Health Risk Behaviours"

_ijerph, 2021, doi:10.3390/ijerph18084220_

Round 1

Reviewer 1 Report

Thank you for the opportunity to review the revised manuscript. The authors have adequately addressed all of the comments I made to the previous draft.

Author Response

Reviewer 1 said that we had adequately addressed all the comments they had made on the previous version of the manuscript.

Reviewer 2 Report

Thank you for addressing reviewer comments in the present version. I have concerns with the new information in this version that describes all three researchers as non-Indigenous. This is concerning as guidance, including the ethical guidance, for Aboriginal and Torres Strait Islander research (by AIATSIS, NHMRC and many more) requires research to be conducted with and for Aboriginal and Torres Strait Islander people. This applies to every stage of the research process, including publication of findings. I suggest that this paper be further revised to include the views as an author, and meaningful input from, at least one Aboriginal and Torres Strait Islander researcher who has been involved in the project or an explanation why this is not possible. 

Author Response

Reviewer 2 said that the revised version had addressed all their reviewer comments. They suggested widening the authorship to include Indigenous co-authorship. Three Indigenous co-authors have reviewed and edited the manuscript, and the manuscript has been amended to reflect this (lines 5-12, lines 261-268, lines 637-640).

Round 2

Reviewer 2 Report

Accept

This manuscript is a resubmission of an earlier submission. The following is a list of the peer review reports and author responses from that submission.

Round 1

Reviewer 1 Report

Manuscript title: Evaluation of B.strong: addressing multiple health risk behaviours through the Queensland Indigenous health worker brief intervention training program

Manuscript ID: ijerph-1063802

The purpose of this paper is to evaluate the efficacy and outcomes of B.strong, an intervention program meant to train health workers to on how to better assist Indigenous clients facing health risks of smoking, poor nutrition and physical inactivity. Overall, it is an interesting paper that I believe will be of interest to readers of this journal. Below are my comments and questions to the authors organized by manuscript section. My comments are lengthy, but it is my hope that they may help improve the paper.

  1. Abstract: Good overview of the study. I just have one question. The abstract states that there was a statistically significant improvement “in the frequency of participants providing client brief interventions in each of the three domains” and then the following sentence reads: “There was a statistically significant improvement in frequency of participants providing brief interventions for multiple health behaviours from pre-workshop to follow-up.” Are these two sentences talking about different findings? It seems to me that they’re saying the same thing.

  1. Introduction: The authors do a good job of contextualizing why the program was needed and how it could help improve the health outcomes of Indigenous Queenslanders. However, it is somewhat confusing to follow and I believe it would benefit from being reorganized. I provide here some suggestions. The paper should ideally move from general to specific, which means that it should start with the statement of the problem, literature on intervention programs, and existing gaps, and then dive into the specific program evaluated in this paper. So, I suggest moving the four last paragraphs of the section (the paragraphs after Box 1) to page 2 before Figure 1. Then, start a new section that focuses specifically on the program being evaluated (perhaps it can be labeled: “Background and structure of the B.strong training program”)

  1. I am not familiar with the terms used for intervention programs and was confused because the authors talk about B.strong being a “brief intervention program” but then state that it qualifies as a “complex intervention” (page 2, line 73 and mentioned again in page 13). Perhaps the authors could clarify for the non-expert readers what a brief intervention is and maybe consider removing the “complex intervention” sentence or explaining what that means.

  1. I love the design of Figure 1. I suggest moving the sentence “As shown in Figure 1, B.strong had multiple components” to the following paragraph before the last sentence (line 91). I also suggest incorporating here the information from the second bullet point in Box 1.

  1. I don’t think Box 1 contributes much to the paper. Rather, it detracts because it’s too wordy and repeats a lot of the information that’s already included in the text. Thus, I suggest removing it and incorporating into the text any information that is not already there.

  1. The paragraph under the heading “Training Content and Delivery” would go better in the previous section than in the methods section (because it’s still talking about the structure of the intervention program, not the evaluation methods)

  1. Data collection: Several questions came up as I was reading this section: Did the pre-, post-, and follow-up surveys ask exactly the same questions? I assume the pre- and post-surveys had a 100% response rate (or maybe not?). What was the response rate of the online follow-up surveys and of the manager/supervisor surveys?

  1. Is there a way for the authors to present the results for pages 8 and 9 in a table or other summarized form? (something perhaps similar to Table 2) There are so many numbers in those paragraphs that it is hard to fully grasp the findings.

  1. Conclusion: Beyond the practical findings about this specific program, how is this paper contributing to the broader literature?

Thank you for the opportunity to read the paper. I hope my comments help to improve the paper.

Reviewer 2 Report

The study "Evaluation of B.strong: addressing multiple health risk behaviours through the Queensland Indigenous health worker brief intervention training program", presents an interesting approach to a strategy for providing educational practices to Indigenous health promoters.
I believe that the study lacks an anthropological focus altogether, and that the strategy designed is not inclusive as it does not take into account the opinion of the indigenous promoters.
The term "clients" should be removed, since it is implied that people pay for the service; this definitely cannot be applied in terms of health promotion.
I believe that this study does not agree with the approaches of the journal.

Reviewer 3 Report

Thank you for the opportunity to review this manuscript. The paper is well written and makes and important contribution to the evidence for Aboriginal and Torres Strait Islander health promotion. I have the following suggestions for improvement:

  1. In the first paragraph, I do not think citing the  decline in smoking for the non-Indigenous population is helpful. 
  2. In the Introduction needs to situate Indigenous health inequity, including inequalities in prevalence of 'behavioural' risk factors in the context of colonisation, racism, inequitable access to the social determinants of health and environmental drivers of smoking/diet/PA. As it stands, the manuscript suggests that Indigenous health inequalities are due to 'unhealthy lifestyle behaviours' (Please reword this term throughout the manuscript as it perpetuates deficit discourse). 
  3. The introduction also needs to include a description of the PHC system in Qld, which is quite different to other Australian states let alone internationally.
  4. Please explain how the training was "culturally appropriate"
  5. Were the SmokeCheck Brief Intervention Training Program and the Aboriginal and Torres Strait Islander Nutrition and  Physical Activity Brief Intervention Program evaluated? The introduction needs to justify why B Strong was based on these. Is there evidence that they were effective and acceptable to Indigenous Queenslanders?
  6. Line 127- the wording "conducted on Indigenous populations" is very unfortunate as it implies Indigenous peoples are passive recipients and are not actively involved in the research process
  7. The introduction is missing a discussion of the evidence for nutrition and physical activity interventions for Indigenous peoples. There have now been several systematic reviews in this field, yet the authors only cite research undertaken with non-Indigenous populations
  8. Much of the Information on the B.Strong training program in the introduction could be moved to the first section of the Methods to keep all the program content /training development information together
  9. Please provide more details of the process of co-design and co-development to ensure that the training and resources were culturally appropriate. What were the roles of the Indigenous Reference Group and Expert Advisory Group? How were Partner and collaborating organisations engaged in program development? How was their input taken on board?
  10. Similarly, please describe any involvement of Indigenous people in the evaluation process. A reflexivity statement from the authors would also be valuable
  11. Has the New World Kirkpatrick Model been used in Indigenous contexts before? Please justify why this framework was appropriate.
  12. Were any objective questions to test participant knowledge included in the survey? Or only perceived/self-assessed knowledge. If the latter, this needs to be made clear throughout the manuscript.
  13. The results of the managers/supervisors survey appear to be under-reported- only two items appear in the results
  14. It seems inappropriate to attribute the increase in Quitline referrals to the B.Strong program if there were also concurrent promotional activities employed by Quitline  to increase referrals for Indigenous clients. Is there other evidence from participant surveys that Quitline referrals increased?
  15. Discussion- There is not sufficient evidence to say that the program has achieved objective 4. The survey feedback, which is worker perceptions re potential benefit, is not an indicator of improved understanding and awareness in Indigenous communities in the longer term and it is misleading to imply that it is.
  16. Line 440- please state that it is perceived knowledge (rather than objectively measured)
  17. Given the decline in some indicators at 3 months (and the workforce turnover that I'm sure exists in the PHC system), please indicate what kind of ongoing training/mentoring/support may be needed to sustain workforce capacity/implementation of program components
  18. Line 481- When Discussing Level 3 it would be more useful to discuss what participants were actually doing differently at three months and whether there was a difference in terms of brief interventions re smoking vs nutrition/physical activity as much of the evidence seems to be about smoking. Can you actually demonstrate from your results that the training significantly changed practice re nutrition or physical activity? 
  19. Since there was no equivalent data to the Quitline referrals for nutrition/physical activity, is this a limitation of the BI model? Is there a lack of programs to which Indigenous clients can be referred for NPA support? This implications of such a gap should be discussed in the limitations section.
  20. The limitations section also needs to address the risk of responder bias/social desirability bias as essentially most of your evaluation data was based on self-report rather than objective measures.
  21. Line 548 Suggest changing "their multiple" to "identified" health risk behaviours
  22. The conclusion that "findings provide strong support for the effectiveness of an Indigenous brief intervention training program addressing multiple health risk behaviours." is overreach. At best the evaluation supports the effectiveness re increasing staff capacity to implement BIs in the workplace. To say health risk behaviours have actually been addressed is misleading.

Round 2

Reviewer 1 Report

Manuscript title: Evaluation of the B.strong Queensland Indigenous health worker brief intervention training program for multiple health risk behaviours

Manuscript ID: ijerph-1063802

Thank you for the opportunity to review the revised manuscript. The authors have made significant revisions to address the concerns of the reviewers during the first round of review. I especially appreciated the reorganization of the manuscript. I think that it now flows better and is easier to follow. I also think that the change to the title gives a better idea to the reader regarding what the paper is about.

I only have one remaining comment, but it is a big one. I am concerned about the very low response rate among managers/supervisors (only n=9; 12% response) and at the 3-month follow-up (only n=142; 13%). I do longitudinal research myself and I completely understand that attrition is a challenge when following up with participants. However, these response rates are well below the accepted threshold. Thus, it is not possible to make reliable conclusions from such responses.

I understand the authors are limited to the available data. My suggestion is to exclude the nine managers from the analyses since this is a very small number of cases to analyze quantitatively. With regards to the 3-month follow-up results, I believe that most of the results are included in the supplementary material, yet the authors still make reference to them in the text. I encourage the authors to add this low follow-up response rate as a limitation in the conclusion and also remind the readers in the text that results must be interpreted with caution given the low response rate.